# Postnatal care following hypertensive disorders of pregnancy: a qualitative study of views and experiences of primary and secondary care clinicians

Debra Bick ,[1] Sergio A Silverio ,[2] Amanda Bye ,[3] Yan-Shing Chang [4]

¹Warwick Clinical Trials Unit, Warwick Medical School, University of Warwick, Coventry, UK
²Department of Women and Children's Health, King's College London, London, UK
³Department of Health Service and Population Research, King's College London, London, UK
⁴Department of Child and Family Health, Florence Nightingale Faculty of Nursing, Midwifery and Palliative Care, King's College London, London, UK

**Correspondence to**
Professor Debra Bick;
debra.bick@warwick.ac.uk

## ABSTRACT

**Objectives** To explore clinicians' views and experiences of caring for postnatal women who had hypertensive disorders of pregnancy (HDP), awareness of relevant National Institute for Health and Care Excellence (NICE) guidance to inform their postnatal management, the extent to which NICE guidance was implemented, barriers and facilitators to implementation and how care could be enhanced to support women's future health.

**Design** A qualitative study using semistructured interviews. Thematic analysis was used for coding and theme generation.

**Setting** Four National Health Service maternity units and three general practice clinics in South-East and South-West London.

**Participants** A maximum variation, purposive sample of 20 clinicians with experience of providing postnatal care to women following HDP.

**Results** Four main themes were generated: variation in knowledge and clinical practice; communication and education; provision of care; locus of responsibility for care. Perceived barriers to implementation of NICE guidance included lack of postnatal care plans and pathways, poor continuity of care, poor antihypertensive medication management, uncertainty around responsibility for postnatal care and women's lack of awareness of the importance of postnatal follow-up for their future health. Some clinicians considered that women were discharged from inpatient care too soon, as primary care clinicians did not have specialist knowledge of HDP management. Most clinicians acknowledged the need for better planning, communication and coordination of care across health settings.

**Conclusions** Evidence of longer term consequences for women's health following HDP is accumulating, with potential for NICE guidance to support better outcomes for women if implemented. Clinicians responsible for postnatal care following HDP should ensure that they are familiar with relevant NICE guidance, able to implement recommendations and involve women in decisions about ongoing care and why this is important. The continued low priority and resources allocated to postnatal services will continue to promote missed opportunities to improve outcomes for women, their infants and families.

### Strengths and limitations of this study

► This study is the first to explore the views and experiences of UK clinicians who provided postnatal care of women who experienced hypertensive disorders in pregnancy, a neglected aspect of maternity care.
► This study reflects increasing evidence that more women are commencing pregnancy at increased risk of developing hypertensive disorders in pregnancy due to older maternal age, higher body mass indexes and other medical complications.
► This study included midwives, obstetricians, general practitioners, obstetric physicians and a health visitor, reflecting a range of clinical practice across secondary and primary care settings.
► The study setting included inner city and urban areas in the south of England. It is possible that clinician experiences in other parts of the UK could reflect differences in populations of pregnant and postnatal women.

## INTRODUCTION

Hypertensive disorders of pregnancy (HDP) include gestational hypertension, chronic hypertension, pre-eclampsia–eclampsia and pre-eclampsia imposed on chronic hypertension. HDP are among the most severe health problems experienced by women during, and in some cases, following pregnancy, and a leading cause of direct maternal deaths globally.[1] Although there is no international consensus on diagnostic criteria,[2] HDP affect around 5%–10% of women, with rates increasing due to associations with older maternal age, obesity and diabetes.[3 4]

Much of the literature to date has focused on pregnancy management of HDP,[2 5] but focus is increasing on postnatal management,[2 6 7] reflecting increasing evidence of risk of recurrence of HDP in subsequent pregnancies. An individual participant data (IPD) meta-analysis reported a recurrence rate of HDP of 20.7% based on 22 cohort studies included in the IPD; recurrence manifested as

pre-eclampsia in 13.8% of the studies, gestational hypertension in 8.6%, and haemolysis, elevated liver enzymes and low platelet count syndrome in 0.2%.[8]

Adverse outcomes are not only restricted to the index pregnancy or subsequent pregnancies, with large observational studies and systematic reviews reporting that HDP increase a woman's lifetime risk of cardiovascular disease (CVD) two- to five-fold compared with women who were normotensive.[9–11] An earlier systematic review and meta-analysis also found women who had pre-eclampsia had significantly increased odds of a fatal or diagnosed cerebral vascular accident and hypertension in later life.[12]

Although underlying mechanisms are currently not well understood, given high and persistent levels of maternal morbidity, appropriate and timely clinician input to assess postnatal risk and promote health and well-being following HDP is imperative. Increased awareness of which women may be at risk, together with advice offered to women on self-management of potentially modifiable risk factors such as weight management and tobacco smoking cessation,[3 13] highlighted the potential opportunity for postnatal care to benefit shorter and longer term health.

In the UK, the majority of women access maternity care provided by the National Health Service (NHS). All women, including those who had HDP, will be discharged following birth to routine contacts with a range of clinicians including midwives (for the first 10–14 days postnatally) and health visitors, and advised to make an appointment at 6–8 weeks with their general practitioner (GP; family doctor) at which point they are discharged from maternity care. Some women may be offered a follow-up consultation with their obstetrician and/or another member of the medical team, but this will depend on local clinical arrangements and capacity.[14]

The National Institute for Health and Care Excellence (NICE) aims to improve outcomes for patients in the UK who receive NHS care through publication of guidance and technology appraisals on provision of effective health, public health and social care across a range of areas, including maternity care. This includes guidance on the routine postnatal care all women and their infants should be offered,[14] and guidance for women who had HDP,[3 13] which includes postnatal lifestyle behaviours, breastfeeding considerations and consideration of longer terms risks to a woman's health.

Despite increasing numbers of women diagnosed with HDP, little is known about clinicians' management of their postnatal care. The aim of this study was to explore the extent to which planning and provision of care for women who had HDP reflected the then available NICE guidance on HDP[13] and routine postnatal care.[14]

## METHODS
Study objectives were to explore:
► Clinicians' views and experiences of caring for women who had HDP, including awareness of relevant NICE guidance and the extent to which planning, and provision of care reflected guidance.
► Barriers and facilitators to implementation of NICE recommendations relevant to postnatal management following HDP.
► Views of where service revisions could be introduced to better meet women's needs and improve quality of care, including the role of multidisciplinary team (MDT).

### Setting
Recruitment took place across four London NHS maternity units and three local general practices based in South-East and South-West London. All four units provided specialist maternity care for women with more complex pregnancies and/or whose infants required neonatal intensive care, in addition to women classed as having a low-risk pregnancy. Three of the maternity units were tertiary centres, with specialist services to treat women and/or infants with the most severe complications of pregnancy and birth and were referral centres for units without these services. The fourth unit was a district general hospital. The general practices were all based in the same locale as three of the maternity units and offered a range of medical services for the local general population, including women living in the hospital catchment area. All of the study sites provided care to ethnically and culturally diverse populations.

### Recruitment and participants
A maximum variation sampling strategy[15 16] was used to recruit a range of clinicians responsible for providing postnatal care for women who had HDP. This strategy was employed to provide a comprehensive understanding of views and experiences of care offered in-line with the study aim and objectives. Clinicians were identified by local study collaborators as having responsibility for providing postnatal care, in-line with study inclusion criteria. Identified clinicians were contacted by a member of the research team (AB) who explained the purpose of the study and what participation would involve. Those who agreed to participate provided written informed consent prior to the research interview.

### Data collection
The interview schedule was developed by the authors with input from a stakeholder group of local clinicians with expertise in HDP and a patient and public involvement group comprising women who had previously experienced HDP. Interviews were conducted face to face by AB, or if not possible, by telephone, and were intended to be semistructured, but conversational in nature. This approach allowed for standard questions to be asked in all interviews, but permitted topics of interest to be probed further, additional questions to be asked by the interviewer if appropriate, and informed exploration of topics in subsequent interviews. Interviews, which lasted

**Table 1** Study ID number and clinical background of participants

| ID | Role | ID | Role |
|---|---|---|---|
| 1 | General practitioner | 11 | General practitioner |
| 2 | General practitioner | 12 | Consultant obstetrician |
| 3 | General practitioner | 13 | Specialist midwife for hypertension |
| 4 | Consultant obstetrician | 14 | Consultant midwife |
| 5 | Obstetric physician | 15 | Consultant obstetrician |
| 6 | Community midwife | 16 | Research midwife and midwifery matron |
| 7 | Specialist midwife for hypertension | 17 | Consultant obstetrician |
| 8 | Clinical research fellow (obstetric trainee) | 18 | Community midwife |
| 9 | Consultant midwife | 19 | General practitioner |
| 10 | Consultant obstetrician | 20 | Health visitor |

for around 30 min, were transcribed verbatim and transcripts were anonymised and checked for inaccuracies.

## Analysis

Data were managed in NVivo V.12 and analysed using thematic analysis,[17 18] a process characterised by inductive, consultative and open coding of the data.[19] The six-phase process of thematic analysis[17] was followed, which entails familiarisation, generating initial codes, searching for and then reviewing themes, defining and offering names for those themes. All data were coded by AB with 20% of the transcripts independently coded by Y-SC. The coding was reviewed by SAS. The researchers met regularly to discuss and revise the themes and ensure analytical rigour, with any discrepancies discussed to ensure coding and analysis satisfied all perspectives.[20] Themes, centred on a core organising concept to sustain its explanatory power within the data, were developed from the codes[17] following discussions. All researchers agreed on the final themes.

Sample size sufficiency, data adequacy and theme saturation were assessed during the analysis process using existing models of thematic concordance and data quality.[21–23] This meant data were fully saturated with the number of participants recruited, and common themes could be found across the dataset, with no new themes being generated by the time the last transcript was analysed.

## PATIENT AND PUBLIC INVOLVEMENT

Women who had experienced HDP informed the original research questions and study design. These women were not involved in the recruitment, or conduct of this study. We presented the findings and discussed the implications of these through a meeting advertised nationally via the charity action on pre-eclampsia, to women who experienced HDP.

## RESULTS

Thirty-four clinicians were invited to interview, 20 of whom agreed to participate, representing the clinical backgrounds of interest (see table 1). Five GPs; six obstetricians; seven midwives (including case load, consultant and specialist hypertension midwives); one clinical research fellow (obstetric trainee) and one health visitor (a nurse or midwife who has undergone additional training in community public health nursing) were recruited and interviewed.

The analysis generated four main themes: variation in knowledge and clinical practice; communication and education; provision of care; locus of responsibility for care, each with two to four subthemes, which are presented in the following sections.

### Variation in knowledge and clinical practice
#### Clinicians' lack of assessment and knowledge

Clinicians were aware of the importance of identifying postnatal women with HDP, but some acknowledged a general lack of assessment of women's blood pressure (BP) during the postnatal period due to the relatively low incidence of adverse outcomes, and lack of recognition of HDP among non-specialist clinicians.

> I think there are other concerns about undertreated hypertension, which we measure poorly. We probably under measure it and therefore under report it because we will get away with it most of the time, because strokes and eclamptic fits are rare in this population but I think… I think walking around with really high blood pressure is probably not good short-term or long-term. (Participant 4—Consultant Obstetrician)

Concerns were also raised about non-specialists' awareness of side-effects of antihypertensive medication they prescribed, as one of the specialist midwives described:

> The GPs, I'm not sure how much they know… just last week (a woman) had been started on medication called methyldopa [antihypertensive drug] by the GP, which is not recommended postnatally because it makes you very depressed, so that's why she thinks she wasn't feeling herself at all. We immediately stopped that medication. She said that that GP is not her normal GP, it was a locum GP, so maybe they just didn't know, so it happens sometimes, and they may not know. (Participant 7—Specialist Midwife for Hypertension)

The consensus of those interviewed was that more had to be done to educate clinicians, particularly those working in primary care settings. Several interviewees suggested postnatal care should be provided by specialists in hypertensive care, as primary care clinicians may not have capacity or relevant specialist knowledge.

> …we definitely need to re-evaluate the whole postnatal pathway because it's definitely, definitely inadequate……… I think what women actually need is

direct access to a midwife or an obstetrician that they can speak to at any time and get seen at any time… And I mean, obviously, that can't just be any midwife or any doctor, I think it needs to be someone with a specialist interest in hypertension. (Participant 13—Specialist Midwife for Hypertension)

### Clinical decision-making based on BP range

Clinicians offered varied responses when asked what BP parameters they used to manage antihypertensive medication. Some acknowledged that practice varied between clinicians, with a lack of adherence to relevant NICE guidance described.

…within our hypertension in pregnancy guideline we have got a section on postnatal, with a guide on which antihypertensives to prescribe, what schedule of reviews should be and fit with what the NICE advice is [on BP management]. How effectively that's delivered, I think, is quite variable. So, you can have it on paper, but it doesn't necessarily translate well to what people do. (Participant 15—Consultant Obstetrician)

Individual experiences of management of women with HDP shaped the practice of some clinicians. Those with more experience of managing HDP expressed greater confidence in their decisions about postnatal management:

NICE say [follow-up checks at] two weeks and six weeks, but as a practicing clinician and on the shop floor and having opportunity, I'm far more, I would say, aggressive in making sure that we're seeing them more often. I think that those kind of timelines, five days, two weeks and six weeks, you've got missed opportunities and actually, women need more or less, and less is as important as more. I've seen a lady today that I've had to reduce the antihypertensive because nobody is brave enough to do that, even though they've seen two other clinicians. (Participant 5—Obstetric Physician)

The need to regularly review women was viewed as important to ensure antihypertensive medication management was appropriate.

…blood pressure's taken on one day may not be representative of which way they're heading. There may be other factors in the immediate post-birth period. Other variables quite strongly affect blood pressure and therefore your choice of antihypertensive may be appropriate at this moment but may need escalation up or down titration, depending on how factors play out. (Participant 4—Consultant Obstetrician)

### Communication and education
### Communication and support between services

Clinicians described several barriers to effective communication following inpatient discharge, including lack of,

or no direct contact with, clinicians involved in a woman's pregnancy care and lack of continuity of postnatal care provider:

I think communication can be quite tricky at times. I know particularly in our area, communication with the midwives isn't quite as easy as it could be in trying to track down who is the actual person that's been seeing this lady—quite often, it tends to be different midwives. (Participant 20—Health Visitor)

Clinicians were conscious of 'gaps' between care provided by specialists in HDP management in hospital and routine primary care services. They also acknowledged the importance of effective communication between clinicians within and across healthcare settings for planning the timing and transition of a woman's care from secondary to primary care. Some clinicians considered that women were discharged too soon from specialist services to primary care teams and could benefit from longer contact with the secondary care team:

Ultimately, these women need lifelong care and they need to go back to their GP so it's about at what time point we transition them. I think it's too early, at the moment. … I think the transition should be in a much more organised and robust way so that women know that they have a schedule of reviews delivered by the maternity services and then they have a six week check, which actually works quite well and is bedded in. (Participant 15—Consultant Obstetrician)

### Women's engagement with advice and education

Effective provision of information for women was viewed as key to assist their understanding the severity of their disease, and encourage adherence to postnatal management plans, taking prescribed medication as advised in terms of dosage and consistent timing, and engaging with primary care for ongoing postnatal monitoring. It was noted that women needed to engage with the information offered, with some women more able to engage than others for a variety of reasons.

Sometimes it's really easy to give advice and sometimes it's really difficult and I wouldn't say it's in relation to someone's high blood pressure or diabetes or anything like that, it's just down to the woman. Some women are going to listen and some just aren't. (Participant 6—Community Midwife)

Several interviewees referred to the timing of when information was offered could influence how well women engaged with understanding the impacts of HDP, recommending that information should be provided to women at the earliest opportunity and reiterated at every care contact:

It may have all been partly intra-partum. Some of them it'll be only, "Oh, your blood pressure went up actually in labour, and then we want to get the baby

delivered". So sometimes it's quite a short period, and it's a period where lots of other things are going on, and they may not have absorbed all those messages. (Participant 19—General Practitioner)

Point of discharge from inpatient care (women with HDP are generally recommended to birth under the care of the obstetric team)[13] was viewed as a particularly pertinent time to commence a discussion about a woman's postnatal care choices, including options for contraception:

I think all women who are hypertensive at delivery should be given a leaflet where all this primary information is given to them so that they understand that they look long term for contraceptive options. There should be a package that everyone with hypertension in pregnancy should receive at the point of discharge with more information links if she wants more later on… (Participant 12—Consultant Obstetrician)

Others also suggested women needed better antenatal education to enable them to more appropriately psychologically prepare for any potential pregnancy complications, such as HDP.

…I hope that this would prepare them better, that if something happens—if their pregnancy takes a slight deviation from their expectations—then they're better prepared to deal with that, psychologically. I think that would have an impact on how they behave postnatally… (Participant 8—Clinical Research Fellow)

There was also recognition among clinicians that the postnatal period was a time when women often prioritised their newborn infants' health over their own, certainly in the first few months as they adjusted to new family life but also a time when women wanted to feel 'normal' again:

Compliance post-delivery is understandably very poor because they're busy with their babies so who's got time to take tablets three times a day? And they don't feel well on these tablets so don't particularly want to take them. (Participant 10—Consultant Obstetrician)

…women don't always take the drugs that are prescribed to them… so it may be that they honestly forget, maybe that life is just very busy, but sometimes it's they just don't want to take the medication. In pregnancy, that might be because they're worried about the effects on the baby. Once they've had the baby, it's because they want to be normal. They're sick of being 'ill'. They've had the baby; they want to be like everyone else who's had a baby. (Participant 9—Consultant Midwife)

## Provision of care
### Improving care
Some participants considered postnatal referral pathways should be standardised to reduce risk of adverse outcomes if women did not receive prompt care when needed:

There should be a standardised postnatal pattern where this is where the woman goes… how is it okay for a woman to be sitting in a big waiting room with a blood pressure of, you know, more than 160/110, it's ridiculous, you know, you're increasing her risk of having a stroke and what not, or eclamptic fit. (Participant 13—Specialist Midwife for Hypertension)

Other suggestions for improving care included review of risk factors by an appropriate MDT, who could develop a plan for longer term follow-up, to be shared with the woman's GP, alongside information for women on longer term health risks:

If everything is stable, they [women] should be seen again at six weeks where you can discuss future risks. You need to see whether there is a residual disease, whether you need to do more investigations and refer to aligned specialities and arrange for long-term follow-up by the GP and inform them about the long-term cardiovascular risks and contraception. (Participant 12—Consultant Obstetrician)

Barriers to care reflected need for regular monitoring of women's BP, medication review and ensuring information on drug safety was adequate, especially if women were breast feeding:

Before they leave hospital, there are barriers in terms of regular, accurate measurement of blood pressure, provision of effective medication, choice of antihypertensive with regards to breastfeeding, adequacy of information, adequacy of safety data on antihypertensives and breastfeeding. (Participant 4—Consultant Obstetrician)

### Limited capacity effects care provision
Clinicians stressed that a lack of capacity across the maternity services was an important barrier to appropriate and timely care especially in the immediate postnatal period.

I think the staff feel under-resourced, overstretched, not well supported and with a long list of jobs that they're not able to prioritise what's important and what's not so important and then meet the expectations of the mothers……… sometimes the midwife, if they find that woman is hypertensive, they're not able to get the medical staff to come and address that issue because the doctor's based on labour ward or elsewhere. (Participant 8—Clinical Research Fellow)

It was also highlighted that GPs had limited capacity due to their workloads and waiting times for women to get an appointment to see them, to engage effectively with postnatal women, meaning that follow-up contacts were not tailored to need.

I think a lot of the work can be done by general practice, but I think, general practice is struggling and also, it's engagement with the woman. It's both ways.

I think all practices have the ability to do this, but the woman has to engage with them as well. If they're being told they've got to wait a long time for a GP appointment, they won't engage. (Participant 5—Obstetric Physician)

For some clinicians, despite limited capacity, they went 'above-and-beyond' their role especially if a woman's health deteriorated:

I tell them that they can contact me directly… it's just a responsibility I've taken on myself as I don't want them to feel alone postnatally and I don't want them to have to go through A&E. Sometimes I'm not here and they do have to go through A&E but if they call… I will see them, and they won't have to sit in A&E. I'll just do it ad hoc, grab a consultant and just say, "Look at this patient now!" which isn't right, it's not fair. I shouldn't have to do that. (Participant 13—Specialist Midwife for Hypertension)

### Need for medical care

Participants discussed how raised BP, plus adverse blood results could result in a longer inpatient stay following the birth, but women were often unaware their BP could remain unstable postnatally:

…anybody whose blood pressure is 140/190, on two occasions, more than four hours apart, or 150/100 with another factor, such as raised PCR, raised creatinine level, raised ART, low platelets……… I say that, postnatally, we do need to monitor you……… sometimes, the women are under the impression that once they have given birth, their blood pressure will stabilise. I, personally, always say, it's kind of three days before we can see what your blood pressure is doing. So, usually, a hospital admission is round about three days, postpartum. (Participant 16—Research Midwife & Midwifery Matron)

Participants in some cases described how they tried to ensure women who required a longer inpatient stay had a room of their own in an effort to deter women from discharging themselves from hospital:

So quite often, if they've got a prolonged stay, you try and find them a side room, so that at least they have a bit of privacy but it's really difficult to get them a side room because, obviously, there are prioritised women who have had a bereavement or a traumatic labour or something. That's often the difficulty I encounter is trying to get them to stay on the postnatal ward and discharge them safely. A lot of them want to self-discharge beforehand. (Participant 8—Clinical Research Fellow)

### Subsequent pregnancies

The risks to maternal health in subsequent pregnancies were also referred to by the participants.

If it's somebody who's already got a pre-existing problem with hypertension and they were thinking about subsequent pregnancies, then we'd want to make sure that they were on an antihypertensive that was safe in pregnancy in case they were to fall pregnant again. (Participant 1—General Practitioner)

Likewise, interviewees discussed the importance of maintenance of a woman's longer term health after HDP, including counselling women on reducing their risk of HDP in a future pregnancy:

So, we talk about exercise, diet, maintaining normal BMI and having your blood pressure checked, maybe every three, six months, with the GP, just to make sure that they are fit following their pregnancy. I will emphasise that they less likely to have a recurrence of their pre-eclampsia during future pregnancies if their BMI is normal and they don't have any health issues. So, we talk about long term health, off the back of the counselling. (Participant 17—Consultant Obstetrician)

### Locus of responsibility for care
#### Responsibility placed on the clinician

Clinicians based in secondary care settings clearly acknowledged their responsibilities when managing women with HDP, but felt they could better fulfil these if continuity of care tailored to women's individual BP results were implemented.

They are seen on the postnatal ward. I would love to have more continuity, but we don't. By and large I aim to make a postnatal plan of care… but the individualisation of that, it needs to take into account what their blood pressure is postnatally and how rapidly it falls. (Participant 4—Consultant Obstetrician)

This contrasted with those in primary care settings who considered NICE guidance lacked clarity with respect to their clinical responsibilities.

I do think that there is a little bit of a difficulty with knowing what the expectation is of a GP when you've got women who are quite complicated. I think that's something that has definitely come up in our practice, where you're not really sure what the responsibility is or what the frequency with which you should be checking it is, and I don't think that the NICE guidance is so clear on that. (Participant 11—General Practitioner)

The overall feeling from secondary care clinicians was that postnatal women in some circumstances (in this case, women who developed gestational hypertension) were perhaps best managed by them; however, the maternity services did not have capacity or funding for this.

I absolutely think there needs to be a robust handover and a longer-term transition. But, I think, women who developed gestational hypertension are still at

risk of post-partum pre-eclampsia and often very labile for the first couple of weeks and need quite a lot of adjusting and weaning. I think if you're delivering women centred care, this should happen alongside assessing her well-being, her baby and I don't think, necessarily, the GP is best placed to do that. I think that would be a much more robust system, rather than this, "Oh you've had your baby"; and immediately you transition back and go and see your GP. (Participant 15—Consultant Obstetrician)

Furthermore, clinicians discussed that tailoring postnatal care to an individual woman could support appropriate clinical decision-making, as getting to know the woman would enable the clinician (in this case a midwife) to note changes in her postnatal health and well-being:

Tailor that care and know the women… from that, you're in a key position as a midwife to make real clinical decisions because you know them, you know something's not right……… You can talk to them about things and you just see a change in them, you pick up stuff about them that might have been lost by a different midwife if every time if they're just doing a urine [test] and blood pressure and then off you go. (Participant 14—Consultant Midwife)

### Responsibility placed on the woman
Some of the clinicians felt the woman also had to take responsibility for her health following HDP, particularly with respect to monitoring of their BP.

Yes, I suppose they would be seen in the clinic if the woman came here for an appointment to discuss [her hypertension]. Otherwise, I haven't personally been involved with any cases. (Participant 3—General Practitioner)

They've got to think about contraception, and long term if they have got a high BMI, exercising, eating sensibly, all these sorts of things. Then they'll have a follow-up with their blood pressure more frequently. I usually say to them if they've had it, that they should have their blood pressure checked more frequently. (Participant 18—Community Midwife)

There was also the recognition that it was the woman, not the GP, who would arrange postnatal follow-up. One of the GPs felt women would only be seen if they (the woman) took the initiative and made an appointment to see their GP.

Usually, the women themselves have known and they've been told to come and see their GP after twoweeks for a blood pressure check……… Postnatal care is very rare, I should think, one in the last twoyears where they've come to have their blood pressure monitor and still on pills. (Participant 2—General Practitioner)

## Discussion
This is the first study to explore in-depth the views and experiences of a group of UK clinicians' on postnatal management of women who had HDP, their awareness of relevant NICE guidance and barriers and facilitators to implementation of NICE recommendations.

Several important barriers to implementation of NICE guidance were highlighted including lack of postnatal care plans and pathways, lack of continuity, poor compliance with antihypertensive medication management, uncertainty around responsibility for postnatal care and in some cases, women's lack of awareness of the importance of postnatal monitoring and follow-up. Views of current routine postnatal provision led some clinicians based in secondary care to consider that women in some circumstances should remain under their care for longer, rather than be discharged to non-specialist 'routine' primary care services. Clinicians acknowledged the need for better planning, increased awareness of NICE guidance, better coordination of care across health settings, better and timely information for women and a greater role for MDT planning.

More clinicians will be required to plan and implement guidance to support the management of increasing numbers of women likely to experience HDP.[3 4] For all clinicians, awareness that HDP can persist beyond the birth, knowledge of signs and symptoms of *de novo* onset, and the sharing of relevant information with women and their families[14 24] could make a significant difference to outcome.

Findings indicate that the postnatal period remains a neglected aspect of a woman's maternity care, despite increasing evidence of widespread and chronic physical and psychological maternal morbidity.[25] Women in the UK are still more likely to die following birth than during pregnancy[26]; however, maternal deaths as a result of HDP declined considerably between 2009 and 2011 (11 women died, 0.42 per 100000 maternities, 95% CI 0.21 to 0.76) and 2012 and 2014 (three women died, 0.11 per 100000 maternities, 95% CI 0.02 to 0.34)[27]; this reduction attributed to implementation of research, audit and evidence-based guidance.

While the Mothers and Babies: Reducing Risk through Audits and Confidential Enquiries (MBRRACE) (UK) findings are very reassuring, none of the opportunities highlighted to further improve care referred to the postnatal period and beyond, despite 11 of the 14 maternal deaths between 2009 and 2014 occurring between 1 and 42days of giving birth.[27] With evidence of risks of HDP on a woman's future cardiovascular health,[4] risk of reoccurrence of HDPs in a future pregnancy,[11] and evidence that around a third of eclampsia cases and more than half of strokes associated with severe pre-eclampsia occur postnatally,[6 28] postnatal care cannot be ignored.

The variation in content of care and lack of awareness of antihypertensive medication as recommended by NICE[3 13] found in the current study supports a small online survey of UK clinicians of antihypertensive adjustment

postnatally, which aimed to explore prescribing clinicians' approaches to down titration of antihypertensive medication following birth.[2] Of the 101/390 (26%) secondary and primary care clinicians approached who completed the survey, labetalol was the the most common antihypertensive drug prescribed. Most clinicians reported adhering to national guidelines when reducing but not increasing women's medication, with wide variation in practice.

The same research team subsequently published results of a feasibility trial of women's self-management of BP.[29] Ninety-one (49%) of 186 women approached from 5 NHS units who had gestational hypertension or pre-eclampsia which required postnatal antihypertensive management were recruited and randomised to self-management (n=45) or usual care (n=46). The self-management intervention, which commenced following inpatient discharge, comprised daily home BP monitoring and automated medication reduction via telemonitoring. Feasibility outcomes included recruitment, retention and compliance with follow-up; 90% of women completed follow-up. Intervention women had lower BP, with the largest difference at 6 weeks postnatally. Diastolic BP was significantly lower in the intervention group to 6 months postnatally. Study limitations included that women were mostly white and from middle-class backgrounds; nevertheless, a larger randomised controlled trial with BP as the primary outcome is an important next step.[29]

Robust evidence to support postnatal antihypertensive management is needed. This includes studies which directly compare outcomes of different hypertensive drugs and BP thresholds for medication adjustment,[29] and studies which address why there are disparities in outcomes among women of different ethnic groups, such as the recent finding that women of black ethnicity with chronic hypertension had a significantly greater risk of adverse perinatal outcomes compared with white women.[30]

What is clear based on our findings is that greater awareness of, and compliance with, NICE guidance[3 13 14] could enhance some postnatal outcomes. These include developing postnatal care plans and pathways during pregnancy[13] to inform appropriate timing of transfer from secondary to primary care services and the clinicians who should be involved. Greater involvement of the MDT could be of benefit, with consideration of a postnatal MDT consultation to assess women's health needs going forward, and the development of a care plan which could be shared with the woman's GP.[25] Continuity of midwifery care could be supported postnatally for as long as the woman and midwife felt was necessary, with a coordinating clinician identified (eg, the midwife or GP) to sign-post women for additional medical support as and when required. Information on breast feeding when taking antihypertensive drugs should be offered to all women,[3 13] preferably when medication is first prescribed, to enable women to make informed decisions about infant feeding. Our findings clearly highlight that all clinicians should familiarise themselves with current guidance on recommended antihypertensive medication for women who are or are not breast feeding, even if not the prescribing clinician, to ensure they can advise women appropriately.[3 13]

Some clinicians in our study were advising women on approaches to self-manage their health to reduce future risk of the consequences of HDP on their health, including lifestyle advice. Of concern is that a lack of an appropriate care pathway postnatally could delay treatment, including preconception care planning, to potentially benefit outcomes of subsequent pregnancies. NICE recommends weight management support and other lifestyle behaviours are discussed with women, with advice reflecting women's individual circumstances, and reiterated as needed at each primary care consultation.[3 13 14] Women at risk of future CVD based on their pregnancy history will potentially require more frequent screening beyond the postnatal period, although robust evidence is needed of the benefit of this.[10] What NICE guidance does not currently specifically support is flexibility with respect to the duration of postnatal care an individual woman may require. Current guidance[3 13 14] 'assumes' that all women are 'recovered' from their pregnancy and birth within the 6–8 weeks postnatal period, when this may not be the case. We do not yet have evidence of what model of postnatal care would be of most benefit for women who have medically complex pregnancies, an area which needs urgent attention.[25]

As a qualitative study, generalisability may be limited. The study was undertaken in London and it is likely that clinicians' experiences will differ in terms of the population of women they are likely to care for, the environment of care and range of specialist maternity services they can access. However, there were several strengths. We interviewed a wide range of secondary and primary care-based NHS clinicians based in inner city and urban areas, providing care to women and families from a wide range of ethnicities, sociodemographic and socioeconomic backgrounds. Some clinicians were linked to tertiary NHS maternity units which care for women with the most serious medical complications and as such were more likely to be 'experts' in HDP management and to have greater insight into some of the issues raised.

## Conclusion

Evidence of longer term maternal health consequences following HDP is accumulating. Clinicians should ensure that relevant NICE guidance is implemented, that they are familiar with guidance and that women are involved in decisions about ongoing care and why this is important. The continued low priority accorded to postnatal care means that missed opportunities to improve outcomes for women, their infants and families will continue.

**Acknowledgements** The authors would like to express their sincere thanks to all of the clinicians who agreed to participate in interviews. They would also like to thank members of their steering group, Jane Sandall, Cath Taylor, Lucy Chappell, Kate Bramham, Pippa Oakeshott and Nick Anim. They also thank the women who had experienced HDP who participated in the patient and public involvement activities which informed our study processes and outcomes.

**Contributors** DB developed the original study concept and study design, with the support of Y-SC and AB. AB completed data collection, and AB, Y-SC and SAS were responsible for data analysis. DB had primary responsibility for overseeing the study, and drafted the first version of the manuscript, with the support of SAS. All authors read and approved the final manuscript.

**Funding** This research is supported by the National Institute for Health Research (NIHR) Applied Research Collaboration South London (NIHR ARC South London) at King's College Hospital NHS Foundation Trust. SAS (King's College London) is supported by the NIHR ARC South London at King's College Hospital NHS Foundation Trust.

**Disclaimer** The views expressed are those of the authors and not necessarily those of the NIHR or the Department of Health and Social Care.

**Competing interests** None declared.

**Patient consent for publication** Not required.

**Ethics approval** Ethics approval was obtained from West Midlands-Solihull Research Ethics Committee on 26th January 2017. REC reference 17/WM/0054.

**Provenance and peer review** Not commissioned; externally peer reviewed.

**Data availability statement** Data are available upon reasonable request. The data sets generated and/or analysed as part of this study are not publicly available due to them containing information that could compromise research participant privacy/consent but are available from the corresponding author on reasonable request.

**ORCID iDs**
Debra Bick http://orcid.org/0000-0002-8557-7276
Sergio A Silverio http://orcid.org/0000-0001-7177-3471
Amanda Bye http://orcid.org/0000-0002-9808-4956
Yan-Shing Chang http://orcid.org/0000-0002-9086-4472

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
