## [Reviewer comments · BMJ Open]

ARTICLE DETAILS

TITLE (PROVISIONAL)	Postnatal care following hypertensive disorders of pregnancy: A qualitative study of views and experiences of primary and secondary care clinicians
AUTHORS	Bick, Debra; Silverio, Sergio A; Bye, Amanda; Chang, Yan-Shing

VERSION 1 – REVIEW

REVIEWER	NMTH Crombag KU Leuven Belgium
REVIEW RETURNED	15-Oct-2019

GENERAL COMMENTS	The paper written reflects an interesting and relevant subject. The subject has a potential for an interesting discussion and suggestions for improvement. However I have some concerns on how the paper is written and set-up: P 3, line 17-33: I do not see how these are limitations or strengths of this study Introduction: I am a non-native English speaker, so I might be wrong, but I think some paragraphs would benefit from rewriting: P4, Line 24: but focus is shifting towards....? Line 26: reflecting awarenesses of.... Line 39-51: Adverse outcomes are not only restricted to the index pregnancy or subsequent pregnancies. Large observational studies and systematic reviews reported on an increase of a woman's lifetime risk of cardiovascular disease [CVD]. HDP increases this risk two-to-five-fold compared with women who were normotensive^{9,10,11}. Women who had pre-eclampsia have significantly increased odds of a fatal or diagnosed cerebral vascular accident [CVA] and hypertension in later life¹². Line 55-59: needs re-writing. From line 59: Studies indicated that increased awareness of women at risk, and advice on self-management offered to those women at risk..... P6. Line 3-17 Study objectives come better at the end of the introduction? Methodology seems to be sound
--

	Results: General comment: there is too much text, with the quotes not always illustrating the written results. P8, line 18-23: Wat is meant by a health visitor? P9, line 26. For me this indicates 'under-measurement, not lack of knowledge. Besides a lack of knowledge, I would suggest there is also a lack of continuity in care, and a clear pathway after a woman has left the hospital. So consider an over-arching them: inadequacy of follow-up post natal care (include lack of knowledge and communication and support). P10, line 37-46: this indicates a 'suspected' lack of adherence to NICE, not sure if this indicates clinical decision making P11, line 3-15: to me this indicates also a lack of adherence towards de NICE guidelines. Line 21-33: also refers to need of follow-up (which is apparently different to NICE??) P14 line 21: limited capacity, isn't that provision of care? P15, line 3 'above-and-beyond' their role, isn't that locus of responsibility of care? P16, medicalisation of care also implicates something different for me, for example to medicalize where this is actually not necessary. To keep someone in hospital while this is not necessary. The quotes used seem to me not referring to that, but more to an underestimation of the risks by newly mothers DISCUSSION The discussion starts with a nice summary of the results (to me more clear then the results itself). P21, line 23-36, I don't see the added value of this paragraph P21 line 41, what is meant by MBRRACE-UK? My suggestion would be to take a new look at the results, and form that built a new discussion, focus on found barriers and facilitators and suggestions for changes in the clinical and outpatient pathways post partum for these women. Refer to NICE guideline, why it is possible that it is not used as it should be (references?)
--	--

REVIEWER	Jon Hyett Royal Prince Alfred Hospital Sydney NSW Australia
REVIEW RETURNED	18-Oct-2019

GENERAL COMMENTS	This is a well-constructed and written study that I found very interesting – as an obstetrician working with women who have this pregnancy outcome. The article is stimulating from a clinical perspective as it made me consider means of improving current standards of care.
---

	I have few criticisms: The introduction is too long and many of the themes are repeated in the discussion; parts could be abridged / deleted. It is a shame that you did not include any junior hospital doctors in clinical positions (registrars / other junior trainees) as, in my experience, they will be involved in a substantial part of managing these women and planning their discharge. This limitation should be mentioned in the discussion. Otherwise – it's a well written paper which would be of value to clinicians.
--	---

VERSION 1 – AUTHOR RESPONSE

Reviewer's comments	Author response
The paper written reflects an interesting and relevant subject. The subject has a potential for an interesting discussion and suggestions for improvement. However I have some concerns on how the paper is written and set-up: P 3, line 17-33: I do not see how these are limitations or strengths of this study Introduction: I am a non-native English speaker, so I might be wrong, but I think some paragraphs would benefit from rewriting: P4, Line 24: but focus is shifting towards....? Line 26: reflecting awarenesses of.... Line 39-51: Adverse outcomes are not only restricted to the index pregnancy or subsequent pregnancies. Large observational studies and systematic reviews reported on an increase of a woman's lifetime risk of cardiovascular disease [CVD]. HDP increases this risk two-to-five-fold compared with women who were normotensive^{9,10,11}. Women who had pre-eclampsia have significantly increased odds of a fatal or diagnosed cerebral vascular accident [CVA] and hypertension in later life¹².	Thank you. Thank you. We have revised some of the points included, in line with being a strength or limitation. We consider that 'is increasing' is appropriate in this sentence.

Line 55-59: needs re-writing. From line 59: Studies indicated that increased awareness of women at risk, and advice on self-management offered to those women at risk..... P6. Line 3-17 Study objectives come better at the end of the introduction? Methodology seems to be sound Results: General comment: there is too much text, with the quotes not always illustrating the written results. P8, line 18-23: Wat is meant by a health visitor? P9, line 26. For me this indicates 'under-measurement, not lack of knowledge. Besides a lack of knowledge, I would suggest there is also a lack of continuity in care, and a clear pathway after a woman has left the hospital. So consider an over-arching them: inadequacy of follow-up post natal care (include lack of knowledge and communication and support). P10, line 37-46: this indicates a 'suspected' lack of adherence to NICE, not sure if this indicates clinical decision making P11, line 3-15: to me this indicates also a lack of adherence towards de NICE guidelines. Line 21-33: also refers to need of follow-up (which is apparently different to NICE??) P14 line 21: limited capacity, isn't that provision of care? P15, line 3 'above-and-beyond' their role, isn't that locus of responsibility of care?	We consider that 'reflecting increasing evidence of..' is appropriate in this sentence. We consider that the paragraph as currently written is appropriate. We consider that the paragraph as currently written is appropriate. We consider that the paragraph as currently written is appropriate. Thank you for this suggestion. We did consider it, but we considered that these were better placed in the Methods section as currently set out. Thank you
---	---

P16, medicalisation of care also implicates something different for me, for example to medicalize where this is actually not necessary. To keep someone in hospital while this is not necessary. The quotes used seem to me not referring to that, but more to an underestimation of the risks by newly mothers DISCUSSION The discussion starts with a nice summary of the results (to me more clear than the results itself). P21, line 23-36, I don't see the added value of this paragraph P21 line 41, what is meant by MBRRACE-UK? My suggestion would be to take a new look at the results, and form that built a new discussion, focus on found barriers and facilitators and suggestions for changes in the clinical and outpatient pathways post partum for these women. Refer to NICE guideline, why it is possible that it is not used as it should be (references?)	We have tried to cut text/quotes where appropriate A health visitor is a public health nurse who works in community settings in the UK. We have added an explanation of this in the text. Thank you. We have revised the text to clarify this that this relates to variation in practice and under-measurement of women's blood pressure, more in line with the quote used. We refer under several of the themes to these issues, so felt it was not appropriate to develop an over-arching theme. Thank you for these helpful comments. We have revised the paper accordingly. Thank you for this comment, we have moved this text to the section entitled 'Provision of Care' as on reflection, we felt it was better under this heading. No, as this refers to some clinicians finding additional time to care for women rather than an issue of knowing who was responsible for providing care for these women. Thank you for this comment. We have revised the heading to 'Need for medical care'. Our findings relate more to the need to provide care which was more medicalised (so it was
--	--

	necessary) and women with HDP may have benefitted from remaining in hospital. Thank you. We are making the point that if clinicians are aware of HDP and the postnatal consequences, it might be possible to prevent adverse outcomes through better discussion of the issues with women and their families. This is the organisation which was established to review maternal deaths and severe maternal morbidity in the UK and Ireland. This has been clarified in the text. Thank you. We have considered these comments and revised the paper as appropriate.
Reviewer 2	
This is a well-constructed and written study that I found very interesting – as an obstetrician working with women who have this pregnancy outcome. The article is stimulating from a clinical perspective as it made me consider means of improving current standards of care. I have few criticisms: The introduction is too long and many of the themes are repeated in the discussion; parts could be abridged / deleted. It is a shame that you did not include any junior hospital doctors in clinical positions (registrars / other junior trainees) as, in my experience, they will be involved in a substantial part of managing these women and planning their discharge. This limitation should be mentioned in the discussion. Otherwise – it's a well written paper which would be of value to clinicians.	Thank you We have revised where appropriate.

VERSION 2 – REVIEW

REVIEWER	Neeltje Crombag KU Leuven, Belgium
REVIEW RETURNED	27-Nov-2019
GENERAL COMMENTS	I think the authors have adapted the comments well, results section I much clearer in my opinion.